# Associations of Head and Neck Cancer with Prior Allergic Rhinitis

**DOI:** 10.3390/cancers17061000

**Published:** 2025-03-17

**Authors:** Shih-Han Hung, Tzong-Hann Yang, Herng-Ching Lin, Chin-Shyan Chen

**Affiliations:** 1Department of Otolaryngology, School of Medicine, Taipei Medical University, Taipei 110, Taiwan; d119100002@tmu.edu.tw; 2Department of Otolaryngology, Wan Fang Hospital, Taipei 110, Taiwan; 3International Ph.D. Program in Medicine, College of Medicine, Taipei Medical University, Taipei 110, Taiwan; 4Department of Otorhinolaryngology, Taipei City Hospital, Taipei 106, Taiwan; dad64@tpech.gov.tw; 5Department of Speech, Language and Audiology, National Taipei University of Nursing and Health, Taipei 112, Taiwan; 6Department of Exercise and Health Sciences, University of Taipei, Taipei 100, Taiwan; 7Department of Otorhinolaryngology, School of Medicine, National Yang Ming Chiao Tung University, Taipei 112, Taiwan; 8Research Center of Data Science on Healthcare Industry, College of Management, Taipei Medical University, Taipei 110, Taiwan; 9School of Health Care Administration, College of Management, Taipei Medical University, Taipei 110, Taiwan; henry11111@tmu.edu.tw; 10Research Center of Sleep Medicine, Taipei Medical University Hospital, Taipei 110, Taiwan; 11Department of Economics, National Taipei University, New Taipei City 237, Taiwan

**Keywords:** head and neck cancers, epidemiology, allergic rhinitis, risk factors

## Abstract

This investigation explores the link between allergic rhinitis (AR) and the onset of head and neck cancers (HNC) utilizing a robust, population-based dataset from Taiwan’s Longitudinal Health Insurance Database 2010. In our case-control study, we analyzed data from 14,913 individuals newly diagnosed with HNC and 59,652 controls matched based on propensity scores. Our findings revealed that 20.19% of the participants had a history of AR, with a higher prevalence observed in the HNC group relative to the controls (26.20% vs. 18.70%). The adjusted odds ratio for prior AR in the HNC cohort was 1.559 (95% CI = 1.494–1.627), suggesting a significant association between AR and increased risk of developing HNC, particularly cancers of the nasopharynx, sinonasal cavities, larynx, salivary glands, and thyroid.

## 1. Introduction

Head and neck cancers (HNC), which develop in the upper aerodigestive tract, represent a major global health challenge. Annually, these malignancies are responsible for approximately 900,000 new cases and over 400,000 fatalities worldwide, underscoring their significant impact on public health [1]. The development of HNC is influenced by a complex interplay of factors. Established risk factors include tobacco use, alcohol consumption, human papillomavirus (HPV) infection, and certain occupational exposures [1,2]. Additionally, recent studies are increasingly indicating that chronic inflammatory states may play a crucial role in the pathogenesis of HNC [3].

Allergic rhinitis (AR) is a common chronic inflammatory condition affecting the nasal mucosa, manifesting symptoms including nasal congestion, rhinorrhea, sneezing, and itching [4]. Considering the persistent inflammatory nature of AR, it is hypothesized that this condition could potentially contribute to carcinogenic processes. The underlying mechanisms might involve prolonged inflammation, oxidative stress, and disruptions in immune regulation [4].

While existing research has examined the link between allergic diseases and various cancer types, the specific relationship between AR and HNC is still not well-defined, with studies showing mixed results. Some research indicates a possible protective effect of AR against HNC, which could be attributed to enhanced immune surveillance or lifestyle variations [5]. On the other hand, other research supports a positive correlation, where the persistent inflammatory state associated with AR could potentially facilitate tumorigenesis [6,7]. It is important to note that the head and neck area is especially prone to chronic inflammation due to its frequent exposure to environmental allergens and pathogens.

This study is designed to explore the potential association between HNC and prior AR by leveraging a large-scale, population-based dataset from Taiwan. Utilizing the National Health Insurance (NHI) program, which covers over 99% of the population across all socioeconomic groups and maintains comprehensive medical records for its beneficiaries [8], we aim to assess if a history of AR increases the risk of developing HNC. Our analysis will control for various potential confounders including age, sex, socioeconomic status, and other medical comorbidities.

## 2. Materials and Methods

### 2.1. Database

We employed the Taiwan Longitudinal Health Insurance Database 2010 (LHID2010) for this case-control study. Since 1995, Taiwan has implemented a mandatory single-payer healthcare system through the NHI program. All residents living in Taiwan for more than four months are required to enroll in the NHI, ensuring they receive comprehensive healthcare coverage. This system features low copayments and minimal wait times, providing equitable access to medical services for all citizens, irrespective of socioeconomic status. The NHI program, now approaching its 30-year milestone, serves as a robust foundation for population-based health research.

The LHID2010 database encompasses medical claim files and beneficiary registrations for a representative sample of 2,000,000 individuals enrolled in the NHI program. This dataset includes several specific registries: a registry for board-certified specialists, catastrophic illness patients, contracted specialty services, medical personnel, contracted beds, medical facilities, drug prescriptions, beneficiaries, and medical services. All personal identifiers within the LHID2010 are anonymized to protect patient privacy, making it a valuable resource for medical research. The database provides a solid platform for gathering real-world evidence, making it instrumental for epidemiological studies focused on chronic conditions and treatment efficacy evaluations.

This study was conducted with ethical approval from the Research Ethics Committee of National Taiwan University (approval number 202312EM054), adhering to the standards set forth in the Declaration of Helsinki. As this research involved the use of deidentified administrative data, the requirement for patient informed consent was waived.

### 2.2. Identification of Study Patients

This observational study encompassed 14,913 patients, all 20 years or older, diagnosed with HNCs for the first time between 1 January 2010, and 31 December 2019. The following tumor sites were included in our analysis: oral cavity, oropharynx, larynx, hypopharynx, nasopharynx, sinonasal cavities, salivary glands (major salivary glands only), and thyroid. Diagnoses were classified according to the International Classification of Diseases (ICD) codes for various cancer types as follows: cancers of the oral cavity (ICD-10-CM codes C00, C02—excluding C02.4, C03-C06; ICD-9-CM codes 140, 141—excluding 141.0 and 141.6, 143, 144, and 145—excluding 145.2, 145.3, and 145.5), oropharynx (ICD-10-CM C01, C02.4, C09, C10; ICD-9-CM 141.0, 141.6, 146), larynx (ICD-10-CM C32; ICD-9-CM 161), hypopharynx (ICD-10-CM C12, C13; ICD-9-CM 148), nasopharynx (ICD-10-CM C11; ICD-9-CM 147), sinonasal (ICD-10-CM C31, C30.0), salivary gland (ICD-10-CM C07, C08; ICD-9-CM 142, 160—excluding 160.1), and thyroid (ICD-10-CM C73; ICD-9-CM 193). The index date was set as the date of the first HNC diagnosis for each patient.

To explore the potential link between HNC and prior AR, we extracted controls aged ≥20 years old from the remaining pool of enrollees in the Registry of Beneficiaries using a propensity-score-matching approach. Controls were selected from individuals without a prior HNC diagnosis at the time of matching to minimize bias due to undiagnosed cases later developing HNC. Propensity scores were computed for the 14,913 HNC patients and eligible controls using a logistic regression model. The variables included in the logistic regression model were age, sex, monthly income, geographic location, urbanization level of residence (ranked from level 1, most urbanized, to level 5, least urbanized), and medical comorbidities including diabetes, hypertension, hyperlipidemia, and human papillomavirus (HPV) infection. These variables were selected based on their well-established association with HNC risk, ensuring that the matching process accounted for known confounders without directly adjusting for AR. Each HNC case was matched with four controls using a nearest neighbor random matching algorithm with a caliper width of ±0.01. This matching ratio was chosen to enhance statistical power while maintaining balance between groups. The index date for controls was assigned based on the date of an outpatient visit within the study period. To further mitigate residual confounding, we applied multivariate logistic regression analysis after propensity-score-matching to adjust for any remaining imbalances in covariates. Sensitivity analyses were conducted to compare the results between matched and unmatched datasets, confirming the robustness of our findings. Ultimately, this resulted in a study cohort of 14,913 HNC patients and 59,652 matched controls.

### 2.3. Measures of Outcomes

This research was designed to investigate the potential relationship between HNC and previous diagnoses of AR. We identified cases of AR using the ICD-9-CM code 477.9 and the ICD-10-CM code J30.9. To ensure diagnostic accuracy, inclusion criteria required individuals to have at least two documented AR diagnoses before the index date, with at least one diagnosis confirmed by a board-certified otolaryngologist

### 2.4. Statistical Analysis

All statistical analyses were conducted using the SAS system (SAS System for Windows, version 9.4, SAS Institute, Cary, NC, USA). We assessed the baseline characteristics of the participants, including age, sex, monthly income, geographic location, urbanization level, and medical comorbidities such as diabetes, hypertension, hyperlipidemia, HPV infection, tobacco use disorder, and alcohol-related disorders. Comparisons between HNC patients and controls were made using chi-square tests for categorical variables and t-tests for continuous variables. To investigate the association between HNC and AR, we employed multivariate logistic regression models, adjusting for the covariates mentioned above. The results were expressed as odds ratios (ORs) with their 95% confidence intervals (CIs) to measure the risk of having prior AR among HNC patients in comparison with controls. A two-sided *p* value of less than 0.05 was considered to indicate statistical significance.

## 3. Results

Table 1 presents the sociodemographic characteristics and medical comorbidities of the study participants. The mean age of the case group was 59.22 ± 14.14 years, compared to 59.53 ± 14.48 years for the control group, with a statistically significant *p* value of 0.018. Despite this statistical significance, the small Cohen’s d value of 0.022 suggests a negligible effect size, indicating no meaningful age difference between the groups. Further analysis revealed no statistically significant differences in sex (*p* > 0.999), monthly income (*p* = 0.889), or urbanization level (*p* = 0.592). Additionally, the prevalence of medical comorbidities such as hyperlipidemia, diabetes, hypertension, HPV infection, tobacco use disorder, and alcohol abuse/alcohol dependence syndrome showed no significant differences between the cases and controls, with *p* values all exceeding 0.999. This uniformity highlights the comparability of the two groups across these variables.

Table 2 presents the distribution of AR diagnoses among study participants, showing a significantly higher prevalence of AR in HNC patients compared to controls. Notably, AR was more common in patients with larynx, nasopharynx, sinonasal, salivary gland, and thyroid cancers. In contrast, no significant difference was observed for oropharynx and hypopharynx cancers, suggesting no strong association in these subtypes.

Table 3 illustrates the covariate-adjusted odds ratios (ORs) for previous AR among HNC patients, displaying a significant overall correlation post adjustment for pivotal factors. Moreover, individual multivariate logistic regressions were conducted for each cancer type examined to delve into the connections between different cancer varieties and AR. Exploratory site-specific analyses suggested varying degrees of association between prior AR diagnosis and specific subtypes of HNC. Notably, stronger associations were observed in cancers of the nasopharynx (OR = 2.933, 95% CI = 2.722–3.160), sinonasal region (OR = 3.100, 95% CI = 2.424–3.964), thyroid (OR = 1.566, 95% CI = 1.447–1.693), larynx (OR = 1.537, 95% CI = 1.307–1.807), and salivary glands (OR = 1.470, 95% CI = 1.158–1.865). However, weaker or no significant associations were observed for cancers of the oral cavity (OR = 0.962, 95% CI = 0.889–1.042), oropharynx (OR = 1.046, 95% CI = 0.883–1.239), and hypopharynx (OR = 1.220, 95% CI = 1.035–1.437). These findings highlight potential differences in the strength of association by cancer subtype but must be interpreted with caution due to inherent limitations of the study design, including possible residual confounding and misclassification biases.

## 4. Discussion

This population-based case-control study, utilizing data from Taiwan’s LHID2010, examined the complex link between AR and HNC. Our results indicate a statistically significant correlation, with a prior diagnosis of AR associated with an increased risk of developing HNC, evidenced by an adjusted odds ratio (OR) of 1.559 (95% CI: 1.494–1.627). This association persisted even after comprehensive adjustments for numerous potential confounders including sociodemographic characteristics and medical comorbidities such hypertension, HPV infection, diabetes, hyperlipidemia, tobacco use, and alcohol-related disorders. These findings emphasize the relevance of AR as a potential risk factor for HNC, underscoring the necessity for further research into the mechanisms that might explain this relationship.

One of the standout observations from our research is the distinct variation in the association between AR and HNC at different anatomical sites. Our analysis revealed a marked increase in the odds ratio (OR) for AR among patients with cancers of the larynx (OR: 1.537), nasopharynx (OR: 2.933), sinonasal area (OR: 3.100), salivary glands (OR: 1.470), and thyroid (OR: 1.566). Conversely, no significant associations were observed for cancers of the oral cavity (OR: 0.962) and oropharynx (OR: 1.046). Additionally, the association with hypopharyngeal cancer was significant but comparatively weaker (OR: 1.220). These findings highlight that the influence of AR on the risk of HNC is not uniformly distributed across the head and neck region; rather, it appears to be modulated by the specific anatomical and physiological attributes of each subsite, along with variations in pathogenesis and the degree of exposure to allergens and inflammatory agents.

The particularly strong associations we observed for nasopharyngeal and sinonasal cancers within the context of AR are striking and merit detailed scrutiny. These cancer sites are proximally and continuously exposed to the nasal cavity, the principal area affected by AR inflammation. Chronic exposure to inhaled allergens leads to a sustained release of inflammatory mediators in the nasal cavity, potentially having a more direct and intense impact on the neighboring nasopharynx and sinonasal areas. Such a persistent inflammatory state may foster a milieu that is conducive to malignant transformations specifically in these locations. This hypothesis aligns with existing research which suggests that chronic inflammation can facilitate carcinogenesis through multiple pathways, including the induction of DNA damage, genomic instability, and the stimulation of cell proliferation and angiogenesis [9,10,11,12].

Conversely, the absence of a statistically significant relationship between AR and cancers of the oral cavity or oropharynx suggests that AR-induced inflammation may exert a less significant impact on these regions, or that other etiological influences are more critical. The oral cavity and oropharynx encounter a continuous influx of external stimuli, such as food, drinks, and microorganisms. The robust immune responses and specialized mucosal defenses in these areas may more effectively manage and resolve inflammatory conditions, potentially reducing the carcinogenic potential of AR-related inflammation. Additionally, well-documented risk factors for oral and oropharyngeal cancers, like tobacco use and alcohol consumption, may dominate and obscure the subtler impacts of inflammation due to AR in these sites [1,13]. This indicates a need for further research to dissect the complex interactions between these predominant risk factors and AR, enhancing our understanding of their collective influence on cancer development in the oral and oropharyngeal regions.

Although less pronounced than the associations observed with nasopharyngeal and sinonasal cancers, the link between AR and laryngeal cancer remains significant. It is possible that chronic inflammation and immune dysregulation associated with AR contribute to carcinogenic processes in the laryngeal mucosa, rather than a direct effect of allergen exposure due to anatomical proximity. This chronic inflammatory environment in the larynx, which AR may intensify, could plausibly contribute to the development of laryngeal cancer. The mechanisms potentially involved are akin to those suggested for nasopharyngeal and sinonasal cancers, including DNA damage, compromised DNA repair processes, and the stimulation of cellular proliferation [14,15]. These pathways collectively highlight the broader impact of chronic inflammation, possibly driven by AR, on cancer susceptibility within the upper aerodigestive tract.

The observed correlation between AR and an increased risk of thyroid cancer is particularly intriguing, warranting deeper investigation. Unlike the upper aerodigestive tract, the thyroid gland is not directly exposed to inhaled allergens. Nonetheless, it may be susceptible to the systemic inflammatory effects of AR. Cytokines and chemokines released during allergic reactions could travel through the bloodstream to the thyroid, fostering a pro-inflammatory environment conducive to cancer development. This possibility suggests a broader systemic impact of AR beyond localized sites of allergen exposure. Alternatively, the connection might stem from immune dysregulation associated with AR, which could alter the thyroid’s vulnerability to neoplastic changes. Additionally, it’s crucial to acknowledge potential confounders that were not considered in our study, such as genetic factors or environmental influences that concurrently affect the risk of both AR and thyroid cancer. These complexities underscore the need for comprehensive research to clarify the pathways linking AR to thyroid carcinogenesis [16,17].

The link between AR and HNC appears to be driven by a complex and multifactorial set of mechanisms, centered around chronic inflammation and immune dysregulation, with additional factors potentially playing contributory roles. Chronic inflammation is a recognized factor in the onset of various cancers, and its role in the pathogenesis of HNC is becoming increasingly evident [9,18]. In the case of AR, continuous exposure to allergens leads to a prolonged inflammatory response in the nasal mucosa. This response is marked by the recruitment and activation of various inflammatory cells such as eosinophils, mast cells, and T helper 2 (Th2) lymphocytes [19]. These cells release a range of pro-inflammatory mediators, including cytokines like interleukin (IL)-4, IL-5, IL-13, and tumor necrosis factor-alpha (TNF-α), as well as chemokines, leukotrienes, and reactive oxygen species (ROS) [20,21]. These mediators contribute to the chronic inflammatory state, which may facilitate the carcinogenic process in the head and neck region by promoting cellular proliferation, DNA damage, and impairing the normal mechanisms of cellular repair [22].

The inflammatory mediators released during AR can promote carcinogenesis via multiple interconnected pathways. Firstly, they can generate oxidative stress and consequent DNA damage, which introduces genomic instability and mutations pivotal to the initiation and progression of cancer [23]. This damage can disrupt normal cellular function and regulatory pathways, laying a foundation for malignant transformation. Secondly, these mediators can foster an environment conducive to cancer by promoting cellular proliferation and inhibiting apoptosis, the process by which potentially harmful cells are naturally eliminated. This imbalance favors the survival and expansion of neoplastic cells [24]. Thirdly, inflammatory mediators can drive angiogenesis, the process of forming new blood vessels. This is crucial for tumor growth and metastasis, providing the necessary nutrients and oxygen to rapidly dividing cancer cells and facilitating the spread of cancer throughout the body [25]. Fourthly, these mediators can also alter the immune response. By potentially impairing the body’s anti-tumor immunity, they enable emerging tumor cells to evade immune detection and destruction [26,27]. This immune modulation is critical, as it allows neoplastic cells to thrive unchecked, further contributing to the complexity of cancer pathogenesis related to chronic inflammation from AR.

The chronic inflammatory milieu in AR may establish a microenvironment that fosters tumor growth and progression within the head and neck region. A key mechanism underlying this process involves Th2 cytokines, which are predominant in AR and have been shown to enhance angiogenesis by upregulating vascular endothelial growth factor (VEGF), a crucial mediator of blood vessel formation [25,28]. Increased VEGF expression facilitates the development of new vasculature, which is essential for sustaining rapidly proliferating tumor cells. Additionally, eosinophils, which are highly prevalent in the nasal mucosa of AR patients, serve as a significant source of VEGF and other pro-angiogenic factors [29,30]. These AR-driven angiogenic processes could provide an essential advantage for tumor development by ensuring an adequate supply of oxygen and nutrients, thereby supporting tumor growth, invasion, and potential metastasis.

Furthermore, the persistent inflammatory state in AR may significantly impair immune surveillance, a critical defense mechanism that prevents malignant transformation. Under physiological conditions, the immune system efficiently detects and eliminates abnormal or precancerous cells, thereby inhibiting tumor initiation. However, chronic inflammation disrupts this equilibrium, potentially reprogramming the immune response from an anti-tumor phenotype to a pro-tumorigenic state [31]. Inflammatory mediators predominant in AR, such as cytokines and chemokines, have been shown to suppress the function of cytotoxic T lymphocytes and natural killer (NK) cells, key components of the body’s innate and adaptive anti-tumor defenses [32]. Moreover, chronic inflammation promotes the accumulation of immunosuppressive cell populations, including regulatory T cells (Tregs) and myeloid-derived suppressor cells (MDSCs), which further attenuate anti-tumor immunity and facilitate immune evasion by nascent tumor cells. This shift toward an immunosuppressive tumor microenvironment provides malignant cells with a survival advantage, fostering tumor progression and metastasis [33,34]. These findings suggest that AR-driven immune dysregulation represents a plausible mechanistic link between chronic allergic inflammation and the heightened risk of HNC.

Our study highlights a significant association between allergic rhinitis (AR) and increased risk of nasopharyngeal (NPC) and sinonasal cancers, raising the question of potential interactions with oncoviruses like Epstein-Barr virus (EBV). EBV is a well-established driver of NPC, promoting chronic inflammation, immune evasion, and epithelial transformation. AR-related Th2-skewed immunity, persistent inflammation, and oxidative stress may impair immune surveillance, potentially facilitating EBV reactivation and viral-driven oncogenesis [35]. Additionally, chronic inflammation could increase epithelial permeability, making the mucosa more susceptible to viral persistence and malignant transformation [36]. While our study did not directly assess EBV status, future research incorporating serological EBV markers, viral load assessments, and immune profiling in AR patients could provide valuable insights into the interplay between chronic allergic inflammation and viral-induced carcinogenesis, aiding in targeted cancer prevention strategies.

Our study possesses several notable strengths that enhance the validity and impact of our findings. The large sample size, derived from a comprehensive population-based database, provides substantial statistical power to detect meaningful associations and allows for detailed subgroup analyses. The use of the LHID2010, which contains detailed medical records for a representative sample of 2,000,000 individuals, minimizes selection bias and enhances the generalizability of our results to the Taiwanese population. The extended follow-up period, spanning a decade, allows for the assessment of the long-term effects of AR on HNC risk. Moreover, our study’s ability to adjust for a wide range of potential confounders, including demographic factors, socioeconomic status, and multiple comorbidities, strengthens the causal inference of the observed associations. Examining different HNC subsites provides valuable insights into the potential site-specific effects of AR, highlighting the importance of considering anatomical variations in future research.

However, it is essential to recognize several limitations of our study. First, the diagnosis of HNC and AR was based solely on ICD codes from the Taiwan National Health Insurance Research Database, which may introduce misclassification bias. To mitigate this, we applied stringent inclusion criteria, requiring at least two AR diagnoses, with one confirmed by a board-certified otolaryngologist. However, the absence of direct validation metrics such as sensitivity, specificity, and positive predictive values for these diagnoses remains a limitation. Given our use of Taiwan’s National Health Insurance Database, findings may not be fully generalizable to populations with different genetic backgrounds, healthcare access, or environmental exposures. Second, the LHID2010 database lacks detailed information on the severity, duration, and specific subtypes of AR, which may influence the strength and nature of the association with HNC. Future studies incorporating more granular data on AR phenotypes and endotypes would be valuable. Third, despite adjusting for multiple potential confounders, the possibility of residual confounding due to unmeasured variables remains. Factors such as environmental exposures (e.g., air pollution, occupational hazards), lifestyle behaviors (e.g., dietary patterns), and genetic predispositions may independently influence both allergic rhinitis and head and neck cancer risk yet were not fully accounted for in our analysis.

There are also concerns for reverse causality, where early symptoms of undiagnosed HNC may have led to increased AR diagnoses. Future prospective studies are necessary to elucidate the temporal relationship between AR and HNC. The key limitation of this study is its retrospective, case-control design, which inherently restricts the ability to establish causality. While we identified a significant association between AR and HNC, these findings should be interpreted strictly as correlations rather than direct causal relationships. Although we discussed potential biological mechanisms underlying this association, unmeasured confounding factors may contribute to the observed results. Future prospective cohort studies and mechanistic research are necessary to further elucidate any causal links between AR and HNC.

Despite these limitations, our findings significantly affect cancer prevention and early detection strategies. The observed association between AR and an increased risk of HNC, particularly for nasopharyngeal, sinonasal, laryngeal, salivary gland, and thyroid cancers, suggests that individuals with AR may represent a high-risk population that could benefit from targeted screening and surveillance programs. This is particularly relevant for those with a long history of severe or uncontrolled AR symptoms who may experience more intense and prolonged periods of chronic inflammation. Nevertheless, it is imperative to recognize that the findings in this study should be construed as an observational discovery of the correlation between AR and HNC, rather than a definitive cause-and-effect relationship. It is noteworthy that tobacco smoking has been documented to exacerbate allergic conditions, while chronic ailments such as hypertension and diabetes can also impact immune and inflammatory responses. Tobacco smoking and chronic inflammation could mediate the pathway from AR-induced inflammation to HNC and adjusting for such a variable would introduce mediator adjustment bias, undermining causal inference. Both AR and HNC may potentially stem from shared underlying factors (as shown in Figure 1).

Future research should focus on validating the AR-HNC association through long-term prospective cohort studies that assess AR severity, duration, treatment history, and environmental factors. Mechanistic studies using in vitro and in vivo models are essential to elucidate the biological pathways linking AR-related inflammation to HNC. Investigating inflammatory mediators, such as cytokines and ROS, and their interaction with established HNC risk factors like tobacco use, alcohol consumption, and HPV infection, could provide insights into disease progression. Additionally, identifying biomarkers or genetic factors may help pinpoint high-risk individuals and guide personalized prevention strategies.

## 5. Conclusions

Our study identified a significant overall association between prior AR and increased risk of developing HNC. Exploratory analyses indicated possible variation in the strength of this relationship across different HNC sites, with stronger associations suggested for nasopharynx, sinonasal cavities, larynx, salivary glands, and thyroid cancers, whereas weaker or no clear associations were observed in oral cavity and oropharyngeal cancers. Given the retrospective and observational nature of our study, these site-specific results should be considered exploratory rather than definitive. Future prospective studies incorporating detailed data on AR severity, duration, and management, as well as controlling for additional environmental and genetic factors, are warranted to further investigate these potential associations. Such research could contribute substantially to a better understanding of the role chronic allergic inflammation may play in head and neck carcinogenesis, ultimately informing targeted prevention and early detection strategies.

## Figures and Tables

**Figure 1 cancers-17-01000-f001:**
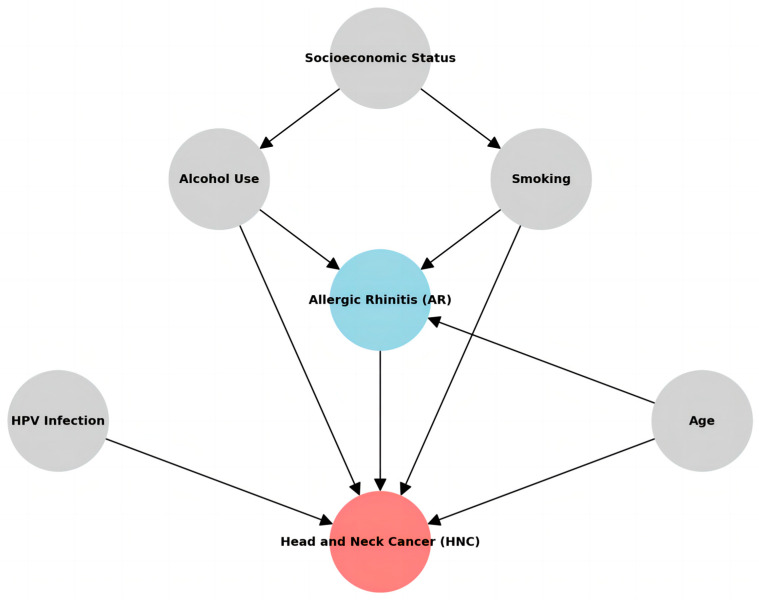
Hypothesized causal relationships among the variables.

**Table 1 cancers-17-01000-t001:** Demographic characteristics and medical comorbidities of patients stratified by HNC.

Variable	Patients with HNC (n = 14,913)		Propensity Score-Matched Controls (n = 59,652)		*p* Value
	Total No.	%	Total No.	%	
Age, mean (SD)	59.22	14.14	59.53	14.48	0.018
Males	9748	65.37	38,992	65.37	>0.999
Monthly Income					0.889
<NT$1~15,841	3491	23.41	14,076	23.60	
NT$15,841~25,000	5364	35.97	21,395	35.87	
≥NT$25,001	6058	40.62	24,181	40.54	
Geographic region					0.033
Northern	6471	43.39	26,408	44.27	
Central	3693	24.76	14,492	24.29	
Southern	4359	29.23	17,389	29.15	
Eastern	390	2.62	1363	2.28	
Urbanization level					0.592
1 (most urbanized)	3858	25.87	15,369	25.76	
2	4176	28.00	17,082	28.64	
3	2709	18.17	10,634	17.83	
4	2063	13.83	8251	13.83	
5 (least urbanized)	2107	14.13	8316	13.94	
Hypertension	6227	41.76	24,908	41.76	>0.999
Hyperlipidemia	5519	37.01	22,076	37.01	>0.999
Diabetes	3577	23.99	14,308	23.99	>0.999
HPV infection	1675	11.23	6700	11.23	>0.999
Tobacco use disorder	883	5.92	3532	5.92	>0.999
Alcohol abuse	449	3.01	1796	3.01	>0.999

Our dataset includes patients from all regions of Taiwan, including affiliated islands such as Penghu.

**Table 2 cancers-17-01000-t002:** Prevalence rates of allergic rhinitis among patients with HNC vs. controls.

Variable	Presence of Allergic Rhinitis		Without Allergic Rhinitis		*p* Value
	n	%	n	%	
Patients with head and neck cancer	3907	26.20	11,006	73.80	<0.001
Patients with oral cavity cancer	825	17.13	3992	82.87	0.007
Patients with oropharynx cancer	169	17.40	802	82.60	0.306
Patients with larynx cancer	206	25.03	617	74.97	<0.001
Patients with hypopharynx cancer	185	19.27	775	80.73	0.650
Patients with nasopharynx cancer	1298	40.07	1941	59.93	<0.001
Patients with sinonasal cancer	110	40.15	164	59.85	<0.001
Patients with salivary gland cancer	94	25.68	272	74.32	0.001
Patients with thyroid cancer	1020	29.45	2443	70.55	<0.001
Controls	11,152	18.70	48,500	81.30	

**Table 3 cancers-17-01000-t003:** Adjusted odds ratio of allergic rhinitis among patients with HNC vs. controls.

Variable	Presence of Allergic Rhinitis
	OR ^a^ (95% CI)
Patients with head and neck cancer	1.559 (1.494~1.627) ***
Patients with oral cavity cancer	0.962 (0.889~1.042)
Patients with oropharynx cancer	1.046 (0.883~1.239)
Patients with larynx cancer	1.537 (1.307~1.807) ***
Patients with hypopharynx cancer	1.220 (1.035~1.437) *
Patients with nasopharynx cancer	2.933 (2.722~3.160) ***
Patients with sinonasal cancer	3.100 (2.424~3.964) ***
Patients with salivary gland cancer	1.470 (1.158~1.865) **
Patients with thyroid cancer	1.566 (1.447~1.693) ***

Notes: ^a^ Adjusted for sex, age, geographic region, monthly income, urbanization level, hypertension, diabetes, hyperlipidemia, alcohol abuse/alcohol dependence syndrome, tobacco use disorder and HPV infection; * *p* < 0.05; ** *p* < 0.01; *** *p* < 0.001.

## Data Availability

Data from the National Health Insurance Research Database, now managed by the Health and Welfare Data Science Center (HWDC), can be obtained by interested researchers through a formal application process addressed to the HWDC, Department of Statistics, Ministry of Health and Welfare, Taiwan (https://dep.mohw.gov.tw/DOS/lp-2506-113.html, accessed on 2 January 2022).

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
