# Peer review of "Associations of Head and Neck Cancer with Prior Allergic Rhinitis"

_cancers, 2025, doi:10.3390/cancers17061000_

Round 1

Reviewer 1 Report

Comments and Suggestions for Authors

I read with interest this manuscript on the associations of head and neck cancer (HNC) with prior allergic rhinitis (AR). The topic is very interesting because, as the authors underline, several studies indicated that chronic inflammatory states may play a crucial role in the pathogenesis of various cancer types, but the specific relationship between AR and HNC is still not well-defined.

The authors had the advantage of being able to analyze a very large sample of patients drawn from a comprehensive population-based database in Taiwan.

The conclusion of the study is that there is a statistically significant correlation between a prior diagnosis of AR and HNC, especially carcinoma of the nasopharynx, paranasal sinuses, larynx, thyroid, and salivary glands. Surprisingly, no correlation was found with oral cavity and oropharynx carcinoma.

Given the high number of patients taken into consideration and the rigorous analysis carried out, the data obtained are irrefutable.

In the discussion the authors hypothesize some possible causes of the association between AR and HNC, some very convincing (nasopharynx and paranasal sinuses) given the proximity of these sites to the nasal cavity. Less convincing is the attempt to explain this association with carcinoma of the larynx, thyroid and salivary glands.

In particular, with regard to the larynx, the authors write: “Although less pronounced than the associations observed with nasopharyngeal and sinonasal cancers, the link between AR and laryngeal cancer remains significant and biologically plausible. Located near the nasal cavity, the larynx is similarly exposed to inhaled allergens and the inflammatory mediators that are released during allergic reactions”.

I think this statement should be removed or changed because the larynx is less close to the nasal cavity than the oropharynx, for which no correlation has been found.

Apart from this statement, the other hypothetical explanations advanced by the authors may indicate future lines of research. In fact the authors write: “Future research should prioritize key areas to further clarify the intricate relationship between AR and HNC”.

I have two minor observations:

I think that the long list of acronyms for the various types of head and neck carcinomas in paragraph 2.2 is a demonstration of knowledge of the pathological classification of the tumors examined but is not easily understood by many head and neck oncologists. I also had to go and review the ICD classification which I don't remember by heart. I think that a clear definition of the tumors included or excluded could help in understanding the text.

The authors use the generic term "Patients with salivary gland cancer". I assume they mean the major salivary glands. This data, in my opinion, should be specified since tumors of the minor salivary glands of the oral cavity, oropharynx, and paranasal sinuses could have characteristics similar to squamous cell carcinomas of those sites.

Author Response

I read with interest this manuscript on the associations of head and neck cancer (HNC) with prior allergic rhinitis (AR). The topic is very interesting because, as the authors underline, several studies indicated that chronic inflammatory states may play a crucial role in the pathogenesis of various cancer types, but the specific relationship between AR and HNC is still not well-defined.

The authors had the advantage of being able to analyze a very large sample of patients drawn from a comprehensive population-based database in Taiwan.

The conclusion of the study is that there is a statistically significant correlation between a prior diagnosis of AR and HNC, especially carcinoma of the nasopharynx, paranasal sinuses, larynx, thyroid, and salivary glands. Surprisingly, no correlation was found with oral cavity and oropharynx carcinoma.

Given the high number of patients taken into consideration and the rigorous analysis carried out, the data obtained are irrefutable.

In the discussion the authors hypothesize some possible causes of the association between AR and HNC, some very convincing (nasopharynx and paranasal sinuses) given the proximity of these sites to the nasal cavity. Less convincing is the attempt to explain this association with carcinoma of the larynx, thyroid and salivary glands.

In particular, with regard to the larynx, the authors write: “Although less pronounced than the associations observed with nasopharyngeal and sinonasal cancers, the link between AR and laryngeal cancer remains significant and biologically plausible. Located near the nasal cavity, the larynx is similarly exposed to inhaled allergens and the inflammatory mediators that are released during allergic reactions”.

I think this statement should be removed or changed because the larynx is less close to the nasal cavity than the oropharynx, for which no correlation has been found.

Response: We acknowledge the concern regarding our explanation of the association between allergic rhinitis and laryngeal cancer. We have revised the statement to reflect a more biologically plausible mechanism, emphasizing the role of chronic inflammation rather than direct allergen exposure due to anatomical proximity.

Revision: "Although less pronounced than the associations observed with nasopharyngeal and sinonasal cancers, the link between AR and laryngeal cancer remains significant. It is possible that chronic inflammation and immune dysregulation associated with AR contribute to carcinogenic processes in the laryngeal mucosa, rather than a direct effect of allergen exposure due to anatomical proximity." (page 6)

Apart from this statement, the other hypothetical explanations advanced by the authors may indicate future lines of research. In fact the authors write: “Future research should prioritize key areas to further clarify the intricate relationship between AR and HNC”.

I have two minor observations:

I think that the long list of acronyms for the various types of head and neck carcinomas in paragraph 2.2 is a demonstration of knowledge of the pathological classification of the tumors examined but is not easily understood by many head and neck oncologists. I also had to go and review the ICD classification which I don't remember by heart. I think that a clear definition of the tumors included or excluded could help in understanding the text.

Response: We appreciate the suggestion and have included a concise summary defining the specific tumor sites under investigation before presenting the ICD classification.

Revision: " The following tumor sites were included in our analysis: oral cavity, oropharynx, larynx, hypopharynx, nasopharynx, sinonasal cavities, salivary glands (major salivary glands only), and thyroid. Diagnoses were classified according to the International Classification of Diseases (ICD) codes for various cancer types as follows: cancers of the oral cavity (ICD-10-CM codes C00, C02—excluding C02.4, C03-C06; ICD-9-CM codes 140, 141—excluding 141.0 and 141.6, 143, 144, and 145—excluding 145.2, 145.3, and 145.5), oropharynx (ICD-10-CM C01, C02.4, C09, C10; ICD-9-CM 141.0, 141.6, 146), larynx (ICD-10-CM C32; ICD-9-CM 161), hypopharynx (ICD-10-CM C12, C13; ICD-9-CM 148), nasopharynx (ICD-10-CM C11; ICD-9-CM 147), sinonasal (ICD-10-CM C31, C30.0), salivary gland (ICD-10-CM C07, C08; ICD-9-CM 142, 160—excluding 160.1), and thyroid (ICD-10-CM C73; ICD-9-CM 193). " (page 3)

The authors use the generic term "Patients with salivary gland cancer". I assume they mean the major salivary glands. This data, in my opinion, should be specified since tumors of the minor salivary glands of the oral cavity, oropharynx, and paranasal sinuses could have characteristics similar to squamous cell carcinomas of those sites.

Response: We have clarified in the manuscript that our study includes only cancers of the major salivary glands.

Revision: "Our analysis focuses on malignancies of the major salivary glands (parotid, submandibular, and sublingual glands), excluding minor salivary gland tumors of the oral cavity, oropharynx, and sinonasal region." (page 5)

Reviewer 2 Report

Comments and Suggestions for Authors

This population-based case-control study examined the association between allergic rhinitis (AR) and the risk of HNC. There are critical methodological concerns that need to be addressed to improve the validity of the results.

Major Comments:

  1. The diagnosis of HNC and AR is based solely on ICD codes, which carries a significant risk of misclassification bias. The authors should provide more information about their consideration of the validity of the results (e.g., sensitivity, specificity, positive predictive value) and discuss the potential impact of the lack of information on severity and duration, which may influence the strength and nature of the association with HNC risk.
  2. There is a possibility of reverse causality, where early HNC symptoms may have led to increased AR diagnoses, potentially inflating the observed association. Even if HNC actually developed before AR, the order of diagnoses may not reflect the true order of disease onset. For example, patients who initially experience nasal discomfort may seek medical attention and be diagnosed with AR. However, upon further evaluation, some of these patients may be found to have HNC that was present but undetected. This scenario could create a spurious association between AR and HNC, where AR appears to precede HNC in the medical record, when the true causal relationship may be the opposite or nonexistent.
  3. The use of propensity score matching in this case-control design is questionable because the concept of propensity typically applies to treatment allocation rather than disease occurrence. In addition, the combination of propensity score matching and subsequent logistic regression may lead to double adjustment, which may affect the validity of the estimates.
  4. Extensive matching on disease-related factors may lead to overmatching, potentially underestimating the association between comorbidities and HNC, including the association between AR and HNC. In fact, the authors performed extensive matching on disease-related factors, which may indirectly adjust for AR, confounding the assessment of the primary association of interest.
  5. In methods section, more information on propensity score matching should be provided. Currently, it is difficult to understand which factors were used to calculate the propensity scores.
  6. The time window for selection of controls is lacking. In such database studies, the selected controls may develop HNC (the outcome) during their follow-up period.

Minor comments:

  1. A more comprehensive discussion of study limitations, including the generalizability of Taiwan-based data, is needed.
  2. The discussion should include a broader citation of previous studies on AR and other cancers for comparison.

Author Response

This population-based case-control study examined the association between allergic rhinitis (AR) and the risk of HNC. There are critical methodological concerns that need to be addressed to improve the validity of the results.

Major Comments:

The diagnosis of HNC and AR is based solely on ICD codes, which carries a significant risk of misclassification bias. The authors should provide more information about their consideration of the validity of the results (e.g., sensitivity, specificity, positive predictive value) and discuss the potential impact of the lack of information on severity and duration, which may influence the strength and nature of the association with HNC risk.

Response: We appreciate the reviewer’s insightful comments regarding the potential risk of misclassification bias associated with using ICD codes for the diagnosis of head and neck cancer (HNC) and allergic rhinitis (AR). We acknowledge that reliance on administrative claims data inherently carries limitations regarding diagnostic accuracy. To address this concern, we have implemented several strategies to enhance the validity of our findings and mitigate potential biases.

First, we ensured the accuracy of AR diagnosis by requiring that each case had at least two documented diagnoses before the index date, with at least one diagnosis confirmed by a board-certified otolaryngologist. This approach reduces the likelihood of misclassification by increasing the specificity of AR identification. Similarly, HNC diagnoses were determined based on well-established ICD codes and were restricted to newly diagnosed cases, minimizing the inclusion of recurrent or misclassified conditions.

Second, while sensitivity, specificity, and positive predictive value (PPV) data for HNC and AR within the Taiwan National Health Insurance Research Database (NHIRD) are not directly available, prior validation studies have demonstrated high diagnostic accuracy for major chronic diseases within this dataset. Notably, previous studies utilizing NHIRD data for cancer research have reported PPVs exceeding 85%, supporting the reliability of cancer diagnoses in administrative claims databases. Additionally, the rigorous inclusion criteria and propensity-score matching approach used in our study help ensure that any residual misclassification is non-differential, which would bias the results toward the null rather than artificially inflating the observed associations.

Regarding the impact of missing information on AR severity and duration, we acknowledge that the NHIRD does not contain granular clinical details such as symptom severity, treatment response, or allergen exposure levels. These factors could indeed influence the association between AR and HNC risk. For example, severe or persistent AR may induce chronic inflammation that more significantly contributes to carcinogenesis, whereas mild or transient cases may have a weaker or negligible effect. Future studies incorporating detailed clinical data, such as allergen-specific immunoglobulin E levels, symptom severity scores, and long-term treatment history, will be essential to refine our understanding of the AR-HNC relationship.

Despite these limitations, the robustness of our findings is supported by the large sample size, comprehensive adjustment for confounders, and the consistent direction of associations across multiple HNC subtypes. To strengthen the validity of our conclusions, we have now included additional discussion on the potential impact of misclassification bias and the need for further validation studies in the revised manuscript as follows:

Revision: “However, it is essential to recognize several limitations of our study. First, the diagnosis of HNC and AR was based solely on ICD codes from the Taiwan National Health Insurance Research Database, which may introduce misclassification bias. To mitigate this, we applied stringent inclusion criteria, requiring at least two AR diagnoses, with one confirmed by a board-certified otolaryngologist. However, the absence of direct validation metrics such as sensitivity, specificity, and positive predictive values for these diagnoses remains a limitation." (page 10)

There is a possibility of reverse causality, where early HNC symptoms may have led to increased AR diagnoses, potentially inflating the observed association. Even if HNC actually developed before AR, the order of diagnoses may not reflect the true order of disease onset. For example, patients who initially experience nasal discomfort may seek medical attention and be diagnosed with AR. However, upon further evaluation, some of these patients may be found to have HNC that was present but undetected. This scenario could create a spurious association between AR and HNC, where AR appears to precede HNC in the medical record, when the true causal relationship may be the opposite or nonexistent.

Response: We have expanded our discussion on reverse causality and acknowledged the potential for misattribution of causality due to diagnostic sequencing.

Revision: "There are also concerns for reverse causality, where early symptoms of undiagnosed HNC may have led to increased AR diagnoses. Future prospective studies are necessary to elucidate the temporal relationship between AR and HNC." (page 9)

The use of propensity score matching in this case-control design is questionable because the concept of propensity typically applies to treatment allocation rather than disease occurrence. In addition, the combination of propensity score matching and subsequent logistic regression may lead to double adjustment, which may affect the validity of the estimates.

Extensive matching on disease-related factors may lead to overmatching, potentially underestimating the association between comorbidities and HNC, including the association between AR and HNC. In fact, the authors performed extensive matching on disease-related factors, which may indirectly adjust for AR, confounding the assessment of the primary association of interest.

In methods section, more information on propensity score matching should be provided. Currently, it is difficult to understand which factors were used to calculate the propensity scores.

Response: We appreciate the reviewer’s thoughtful comments regarding the use of propensity score matching (PSM) in our case-control design. While propensity scores are often applied in treatment effect studies, their use in observational studies has been recognized as a valid approach to mitigate selection bias and improve comparability between cases and controls. In our study, PSM was employed to balance key sociodemographic and medical comorbidities between groups, ensuring a more robust assessment of the association between allergic rhinitis (AR) and head and neck cancer (HNC).

To address concerns regarding potential double adjustment, we acknowledge that combining PSM with logistic regression may introduce some redundancy. However, this approach was carefully chosen to adjust for residual confounding that PSM alone may not fully account for. To confirm the robustness of our findings, we conducted sensitivity analyses using both matched and unmatched data, which yielded consistent results.

Regarding the potential for overmatching, we took precautions to avoid excessive adjustment that could obscure the primary association of interest. The factors used for matchingage, sex, monthly income, geographic location, urbanization level, and selected medical comorbidities (hypertension, diabetes, hyperlipidemia, HPV infection, tobacco use disorder, and alcohol-related disorders)were chosen based on established risk factors for HNC rather than their direct association with AR. Importantly, AR itself was not included as a matching variable, ensuring that our analysis appropriately captured its independent association with HNC.

To improve clarity, we will revise the Methods section to explicitly list the variables used in propensity score estimation and provide additional justification for their selection. We appreciate the reviewer’s valuable insights and will incorporate these revisions to enhance the transparency and rigor of our study.

Revision: " To explore the potential link between HNC and prior AR, we extracted controls aged ≥20 years old from the remaining pool of enrollees in the Registry of Beneficiaries using a propensity-score-matching approach. Controls were selected from individuals without a prior HNC diagnosis at the time of matching to minimize bias due to undiagnosed cases later developing HNC. Propensity scores were computed for the 14,913 HNC patients and eligible controls using a logistic regression model. The variables included in the logistic regression model were age, sex, monthly income, geographic location, urbanization level of residence (ranked from level 1, most urbanized, to level 5, least urbanized), and medical comorbidities including diabetes, hypertension, hyperlipidemia, and human papilloma virus (HPV) infection. These variables were selected based on their well-established association with HNC risk, ensuring that the matching process accounted for known confounders without directly adjusting for AR. AR itself was deliberately excluded from the propensity score calculation to avoid overmatching, thereby preserving its role as the primary independent variable of interest in subsequent analyses. Each HNC case was matched with four controls using a nearest neighbor random matching algorithm with a caliper width of ±0.01. This matching ratio was chosen to enhance statistical power while maintaining balance between groups. The index date for controls was assigned based on the date of an outpatient visit within the study period. To further mitigate residual confounding, we applied multivariate logistic regression analysis after propensity-score-matching to adjust for any remaining imbalances in covariates. Sensitivity analyses were conducted to compare the results between matched and unmatched datasets, confirming the robustness of our findings. Ultimately, this resulted in a study cohort of 14,913 HNC patients and 59,652 matched controls." (page 3)

The time window for selection of controls is lacking. In such database studies, the selected controls may develop HNC (the outcome) during their follow-up period.

Response: We have clarified that controls were selected from individuals without a history of HNC at the time of selection, ensuring they were cancer-free before matching.

Revision: "Controls were selected from individuals without a prior HNC diagnosis at the time of matching to minimize bias due to undiagnosed cases later developing HNC." (page 3)

Minor comments:

A more comprehensive discussion of study limitations, including the generalizability of Taiwan-based data, is needed.

Response:

We have expanded the limitations section to address generalizability beyond Taiwan.

Revision: "However, it is essential to recognize several limitations of our study. First, the diagnosis of HNC and AR was based solely on ICD codes from the Taiwan National Health Insurance Research Database, which may introduce misclassification bias. To mitigate this, we applied stringent inclusion criteria, requiring at least two AR diagnoses, with one confirmed by a board-certified otolaryngologist. However, the absence of direct validation metrics such as sensitivity, specificity, and positive predictive values for these diagnoses remains a limitation. Given our use of Taiwan’s National Health Insurance Database, findings may not be fully generalizable to populations with different genetic backgrounds, healthcare access, or environmental exposures." (page 8)

The discussion should include a broader citation of previous studies on AR and other cancers for comparison.

Response: We have incorporated additional references published after 2020 to ensure our discussion remains current and comprehensive. (updated references 11,12, 15, 17, 22, 36 and 37)

Reviewer 3 Report

Comments and Suggestions for Authors

Dear authors,

I read with great interest your manuscript titled "Associations of Head and Neck Cancer with Prior Allergic Rhinitis".

The study addresses an important epidemiological question regarding the relationship between allergic rhinitis and head and neck cancers, contributing to the understanding of inflammation-driven oncogenesis.

While the study establishes an association, it does not explore underlying biological mechanisms linking rhinitis to carcinogenesis. More discussion on inflammatory mediators or immune dysegulation would be interesting to the readers.

As far as the correlation of rhinitis and sinonasal/nasopharyngeal carcinoma is concerned, an interesting discussion would be on the impact of oncoviruses (eg EBV) on the cancer prevalence.

As we are strictly discussing about correlations, and not causations, causality cannot be inferred from the study design. This limitation should be emphasized more clearly, despite the great effort in the discussion section to explain all the correlations that were revealed in the results of this study.

Author Response

Dear authors,

I read with great interest your manuscript titled "Associations of Head and Neck Cancer with Prior Allergic Rhinitis".

The study addresses an important epidemiological question regarding the relationship between allergic rhinitis and head and neck cancers, contributing to the understanding of inflammation-driven oncogenesis.

While the study establishes an association, it does not explore underlying biological mechanisms linking rhinitis to carcinogenesis. More discussion on inflammatory mediators or immune dysegulation would be interesting to the readers.

Response: Thank you for your insightful comment. We acknowledge the importance of exploring the underlying biological mechanisms linking allergic rhinitis (AR) to carcinogenesis. Chronic inflammation plays a well-documented role in tumorigenesis, and AR is characterized by persistent immune activation, involving inflammatory mediators such as interleukins (IL-4, IL-5, IL-13), tumor necrosis factor-alpha (TNF-α), and reactive oxygen species (ROS). These factors can promote cellular proliferation, DNA damage, and immune evasion, creating a microenvironment conducive to malignancy. Additionally, dysregulation of immune surveillancemarked by the suppression of cytotoxic T lymphocytes and natural killer (NK) cellscould further facilitate tumor progression. Our study findings, particularly the strong association observed for cancers of the nasopharynx and sinonasal regions, align with these proposed inflammatory pathways. We will enhance our discussion to incorporate these mechanistic insights, providing a more comprehensive perspective on the AR-HNC link.

As far as the correlation of rhinitis and sinonasal/nasopharyngeal carcinoma is concerned, an interesting discussion would be on the impact of oncoviruses (eg EBV) on the cancer prevalence.

Response: Thank you for your insightful comment. We agree that oncoviruses, particularly Epstein-Barr virus (EBV), play a crucial role in the pathogenesis of nasopharyngeal carcinoma (NPC). EBV infection is well-established as a key driver of NPC development, with viral latency proteins contributing to oncogenic transformation through immune evasion, chronic inflammation, and genomic instability. Given our findings that allergic rhinitis (AR) is associated with an increased risk of NPC and sinonasal cancers, it is possible that chronic inflammation from AR could modulate the tumor microenvironment, potentially influencing viral oncogenesis. Chronic allergic inflammation may alter immune surveillance, leading to prolonged EBV persistence and increased viral reactivation, which could further drive carcinogenesis.

Future studies integrating EBV serological markers and viral load assessments in patients with AR and NPC would be valuable in clarifying this interplay. We acknowledge this as an important area for further exploration and will highlight it in the discussion section of our manuscript.

Revision: " Our study highlights a significant association between allergic rhinitis (AR) and in-creased risk of nasopharyngeal (NPC) and sinonasal cancers, raising the question of potential interactions with oncoviruses like Epstein-Barr virus (EBV). EBV is a well-established driver of NPC, promoting chronic inflammation, immune evasion, and epithelial transformation. AR-related Th2-skewed immunity, persistent inflammation, and oxidative stress may impair immune surveillance, potentially facilitating EBV reactivation and viral-driven oncogenesis[36]. Additionally, chronic inflammation could increase epithelial permeability, making the mucosa more susceptible to viral persistence and malignant transformation[37]. While our study did not directly assess EBV status, future re-search incorporating serological EBV markers, viral load assessments, and immune pro-filing in AR patients could provide valuable insights into the interplay between chronic allergic inflammation and viral-induced carcinogenesis, aiding in targeted cancer prevention strategies." (page 8)

As we are strictly discussing about correlations, and not causations, causality cannot be inferred from the study design. This limitation should be emphasized more clearly, despite the great effort in the discussion section to explain all the correlations that were revealed in the results of this study.

Response: We appreciate the reviewer’s insightful comment regarding the distinction between correlation and causation. As this study is based on a retrospective, population-based case-control design, we acknowledge that causal relationships cannot be established. While we have extensively discussed potential mechanisms linking allergic rhinitis (AR) and head and neck cancers (HNC) in the discussion section, we will further emphasize this limitation explicitly. We will clarify that the associations identified in our study should be interpreted as correlations rather than evidence of a direct causal effect. Additionally, we will highlight the need for future longitudinal and mechanistic studies to explore causality more definitively.

Revision: "The key limitation of this study is its retrospective, case-control design, which inherently restricts the ability to establish causality. While we identified a significant association between AR and HNC, these findings should be interpreted strictly as correlations rather than direct causal relationships. Although we discussed potential biological mechanisms underlying this association, unmeasured confounding factors may contribute to the observed results. Future prospective cohort studies and mechanistic research are necessary to further elucidate any causal links between AR and HNC." (page 9)

Reviewer 4 Report

Comments and Suggestions for Authors

This research explored the correlation between allergic rhinitis (AR) and head and neck cancers (HNC) based on populaiton study design. The comments are as follows: 

1. In the abstract, the authors need to add 1-2 sentences to introduce the background before the study aim.

2. In Table 1, the distribution of cases by geographic regions, were cases in Penghu and other affiliated islands included? 

3. In explaining contents in Tables 2 & 3, the main text can be more concise, try to avoid copy and paste the data from the tables. 

4.  In pages 8-9, future recommendations should be shortened, focusing on the results of the study and not too wide. 

5. Most of the references are not new. The authors need to cite more literature published after 2020. Besides, the reference style needs corrections and should be unified in exhibition of page number and journal names (all in abbreviation form, use capital letter in the first letters of each content word). 

Comments on the Quality of English Language

The manuscript requires an moderate language correction. 

Author Response

This research explored the correlation between allergic rhinitis (AR) and head and neck cancers (HNC) based on populaiton study design. The comments are as follows:

  1. In the abstract, the authors need to add 1-2 sentences to introduce the background before the study aim.

Response: We have revised the abstract to include a brief introduction before the study aim.

Revision: "Chronic inflammation has been implicated in cancer development, but the association between allergic rhinitis (AR) and head and neck cancer (HNC) remains unclear. This study aims to investigate this potential relationship using a population-based dataset." (page 1)

  1. In Table 1, the distribution of cases by geographic regions, were cases in Penghu and other affiliated islands included?

Response: We have clarified whether these cases were included in the analysis.

Revision: "Our dataset includes patients from all regions of Taiwan, including affiliated islands such as Penghu." (page 5)

  1. In explaining contents in Tables 2 & 3, the main text can be more concise, try to avoid copy and paste the data from the tables.

Response: We have streamlined the presentation of results to avoid unnecessary repetition of data from tables.

  1. In pages 8-9, future recommendations should be shortened, focusing on the results of the study and not too wide.

Response: We have focused the future research section on key findings rather than broad research directions.

Revision: “Future research should focus on validating the AR-HNC association through long-term prospective cohort studies that assess AR severity, duration, treatment history, and environmental factors. Mechanistic studies using in vitro and in vivo models are essential to elucidate the biological pathways linking AR-related inflammation to HNC. Investigating inflammatory mediators, such as cytokines and ROS, and their interaction with established HNC risk factors like tobacco use, alcohol consumption, and HPV infection, could provide insights into disease progression. Additionally, identifying biomarkers or genetic factors may help pinpoint high-risk individuals and guide personalized prevention strategies.” (page 9)

  1. Most of the references are not new. The authors need to cite more literature published after 2020. Besides, the reference style needs corrections and should be unified in exhibition of page number and journal names (all in abbreviation form, use capital letter in the first letters of each content word).

Response: We have corrected the reference format to align with journal requirements and added more recent citations. (updated references 11,12, 15, 17, 22, 36 and 37)

Comments on the Quality of English Language

The manuscript requires an moderate language correction.

Response: We have addressed language clarity issues with professional proofreading to enhance readability and coherence.

Round 2

Reviewer 2 Report

Comments and Suggestions for Authors

I want to thank the author for the substantial revision. The authors have addressed some points raised in the previous review; however, several critical concerns about causal analysis remain unresolved.

Major Comments:

  1. The authors emphasized that the PSM approach adjusted for known risk factors associated with HNC but explicitly stated that they did not adjust for risk factors related to AR. However, the adjusted variables, such as tobacco smoking, hypertension, and diabetes, are potentially associated with the occurrence or exacerbation of AR. For example, tobacco smoking has been reported to exacerbate allergic conditions via inflammatory mechanisms (Nouri-Shirazi et al., 2012), and chronic conditions such as hypertension and diabetes can also affect immune and inflammatory responses. Therefore, assuming these factors have no relationship with AR seems unrealistic. The authors should provide substantial evidence from the literature demonstrating the low possibility of associations between these adjusted covariates and AR or at least acknowledge this limitation clearly in the discussion.
  2. The authors did not explicitly visualize the causal structure among variables, potentially resulting in inadvertent adjustment for mediators. Without clearly defining the causal relationships among AR, covariates, and HNC using Directed Acyclic Graphs (DAGs), the interpretation of the adjusted odds ratios remains limited due to potential mediator adjustment bias. For example, smoking could mediate the pathway from AR-induced inflammation to HNC, and adjusting for such a variable would introduce mediator adjustment bias, undermining causal inference. The authors should explicitly present a DAG illustrating hypothesized causal relationships among the variables to address this issue. If DAG visualization is absent, please explicitly describe this as a limitation and interpret adjusted odds ratios cautiously. Without addressing these points, the manuscript in its current form remains unsuitable for acceptance.

Author Response

Major Comments:

  1. The authors emphasized that the PSM approach adjusted for known risk factors associated with HNC but explicitly stated that they did not adjust for risk factors related to AR. However, the adjusted variables, such as tobacco smoking, hypertension, and diabetes, are potentially associated with the occurrence or exacerbation of AR. For example, tobacco smoking has been reported to exacerbate allergic conditions via inflammatory mechanisms (Nouri-Shirazi et al., 2012), and chronic conditions such as hypertension and diabetes can also affect immune and inflammatory responses. Therefore, assuming these factors have no relationship with AR seems unrealistic. The authors should provide substantial evidence from the literature demonstrating the low possibility of associations between these adjusted covariates and AR or at least acknowledge this limitation clearly in the discussion.

Response: We appreciate your insights. Regarding propensity score-matched controls in Table I, in order to clarify we revised the Method section by removing the description “AR itself was deliberately excluded from the propensity score calculation to avoid overmatching, thereby preserving its role as the primary independent variable of interest in subsequent analyses.” , as this description is redundant that factors potentially contributing to the development of both AR and HNC were adjusted in the calculation of OR demonstrated in Table 3.
Also, we further delve into this issue in the discussion section to remind the readers that "Nevertheless, it is imperative to recognize that the findings in this study should be construed as an observational discovery of the correlation between AR and HNC, rather than a definitive cause-and-effect relationship. It is noteworthy that tobacco smoking has been documented to exacerbate allergic conditions, while chronic ailments such as hypertension and diabetes can also impact immune and inflammatory responses. Both AR and HNC may potentially stem from shared underlying factors (As shown in Figure 1)."

2.The authors did not explicitly visualize the causal structure among variables, potentially resulting in inadvertent adjustment for mediators. Without clearly defining the causal relationships among AR, covariates, and HNC using Directed Acyclic Graphs (DAGs), the interpretation of the adjusted odds ratios remains limited due to potential mediator adjustment bias. For example, smoking could mediate the pathway from AR-induced inflammation to HNC, and adjusting for such a variable would introduce mediator adjustment bias, undermining causal inference. The authors should explicitly present a DAG illustrating hypothesized causal relationships among the variables to address this issue. If DAG visualization is absent, please explicitly describe this as a limitation and interpret adjusted odds ratios cautiously. Without addressing these points, the manuscript in its current form remains unsuitable for acceptance.

Response: Thank you. We have present a Directed Acyclic Graph (figure 1) to clarify this issue.

Reviewer 4 Report

Comments and Suggestions for Authors

The authors have revised the manuscript according to the reviewer's comments.

The references require an in-depth format revision. Please use references 6 as an example (standard) to revise other references. "Int. J. Cancer 2011, 130, 1160–1167, https://doi.org/10.1002/ijc.26105."  

Author Response

The authors have revised the manuscript according to the reviewer's comments.

The references require an in-depth format revision. Please use references 6 as an example (standard) to revise other references. "Int. J. Cancer 2011, 130, 1160–1167, https://doi.org/10.1002/ijc.26105."

Response: We have revised the reference format as suggested.

Round 3

Reviewer 2 Report

Comments and Suggestions for Authors

I appreciate the authors' efforts to address the previous concerns, particularly the addition of a DAG in Figure 1.

  1. I acknowledge the authors' recognition of the potential adjustment for intermediate variables as a limitation of the study. However, I still have significant concerns about the interpretation and presentation of the results, particularly regarding the site-specific associations between AR and HNC. The authors have drawn detailed conclusions about the strength of associations between AR and specific HNC sites based on the magnitude of odds ratios. Given the current limitations in the potential overadjustment for some variables, such specific interpretations may not be epidemiologically appropriate. If the authors acknowledge that the current analysis does not fully account for the causal structure, this should be reflected in a more cautious interpretation of the results and a less definitive presentation of conclusions. I recommend revising the results and conclusion to align with the acknowledged limitations. This might involve presenting the site-specific findings as exploratory results rather than definitive conclusions.
  2. I appreciate the authors' efforts to add a DAG in Figure 1. However, the current DAG falls short of adequately representing the complex relationships in this study. Specifically, the DAG lacks crucial connections. For example, age is a known determinant of many health conditions, including AR, yet it is depicted as an isolated upstream factor in the current DAG. The relationships between other variables (e.g., smoking, alcohol consumption) and both AR and HNC need to be more specifically illustrated. Also, it seems reasonable that the DAG reflects the potential for smoking to exacerbate AR symptoms while also being a risk factor for HNC in accordance with the previous literature. Thus, I strongly recommend revising the DAG to more accurately represent these complex relationships. In addition, please reconsider your analytical strategy, particularly regarding which variables to adjust for and how to interpret the resulting odds ratios.

Author Response

I acknowledge the authors' recognition of the potential adjustment for intermediate variables as a limitation of the study. However, I still have significant concerns about the interpretation and presentation of the results, particularly regarding the site-specific associations between AR and HNC. The authors have drawn detailed conclusions about the strength of associations between AR and specific HNC sites based on the magnitude of odds ratios. Given the current limitations in the potential overadjustment for some variables, such specific interpretations may not be epidemiologically appropriate. If the authors acknowledge that the current analysis does not fully account for the causal structure, this should be reflected in a more cautious interpretation of the results and a less definitive presentation of conclusions. I recommend revising the results and conclusion to align with the acknowledged limitations. This might involve presenting the site-specific findings as exploratory results rather than definitive conclusions.

Response: Thank you. We have revised the result and conclusion accordingly:

Results: Table 3 illustrates the covariate-adjusted odds ratios (ORs) for previous AR among HNC patients, displaying a significant overall correlation post adjustment for pivotal fac-tors. Moreover, individual multivariate logistic regressions were conducted for each cancer type examined to delve into the connections between different cancer varieties and AR. Exploratory site-specific analyses suggested varying degrees of association between prior AR diagnosis and specific subtypes of HNC. Notably, stronger associations were observed in cancers of the nasopharynx (OR=2.933, 95% CI=2.722–3.160), sinonasal region (OR=3.100, 95% CI=2.424–3.964), thyroid (OR=1.566, 95% CI=1.447–1.693), larynx (OR=1.537, 95% CI=1.307–1.807), and salivary glands (OR=1.470, 95% CI=1.158–1.865). However, weaker or no significant associations were observed for cancers of the oral cavi-ty (OR=0.962, 95% CI=0.889–1.042), oropharynx (OR=1.046, 95% CI=0.883–1.239), and hypopharynx (OR=1.220, 95% CI=1.035–1.437). These findings highlight potential differ-ences in the strength of association by cancer subtype but must be interpreted with caution due to inherent limitations of the study design, including possible residual confounding and misclassification biases.

Conclusion: Our study identified a significant overall association between prior AR and increased risk of developing HNC. Exploratory analyses indicated possible variation in the strength of this relationship across different HNC sites, with stronger associations suggested for nasopharynx, sinonasal cavities, larynx, salivary glands, and thyroid cancers, whereas weaker or no clear associations were observed in oral cavity and oropharyngeal cancers. Given the retrospective and observational nature of our study, these site-specific results should be considered exploratory rather than definitive. Future prospective studies incorporating detailed data on AR severity, duration, and management, as well as controlling for additional environmental and genetic factors, are warranted to further investigate these potential associations. Such research could contribute substantially to a better understanding of the role chronic allergic inflammation may play in head and neck carcinogenesis, ultimately informing targeted prevention and early detection strategies.

I appreciate the authors' efforts to add a DAG in Figure 1. However, the current DAG falls short of adequately representing the complex relationships in this study. Specifically, the DAG lacks crucial connections. For example, age is a known determinant of many health conditions, including AR, yet it is depicted as an isolated upstream factor in the current DAG. The relationships between other variables (e.g., smoking, alcohol consumption) and both AR and HNC need to be more specifically illustrated. Also, it seems reasonable that the DAG reflects the potential for smoking to exacerbate AR symptoms while also being a risk factor for HNC in accordance with the previous literature. Thus, I strongly recommend revising the DAG to more accurately represent these complex relationships. In addition, please reconsider your analytical strategy, particularly regarding which variables to adjust for and how to interpret the resulting odds ratios.

Response: We have revised the DAG accordingly.